# Crop switching reduces agricultural losses from climate change in the United States by half under RCP 8.5

James Rising [1✉] & Naresh Devineni [2]

A key strategy for agriculture to adapt to climate change is by switching crops and relocating crop production. We develop an approach to estimate the economic potential of crop reallocation using a Bayesian hierarchical model of yields. We apply the model to six crops in the United States, and show that it outperforms traditional empirical models under cross-validation. The fitted model parameters provide evidence of considerable existing climate adaptation across counties. If crop locations are held constant in the future, total agriculture profits for the six crops will drop by 31% for the temperature patterns of 2070 under RCP 8.5. When crop lands are reallocated to avoid yield decreases and take advantage of yield increases, half of these losses are avoided (16% loss), but 57% of counties are allocated crops different from those currently planted. Our results provide a framework for identifying crop adaptation opportunities, but suggest limits to their potential.

[1] Grantham Research Institute, London School of Economics, London WC2A 2AE, UK. [2] Department of Civil Engineering, City University of New York (City College), New York, NY 10031, USA. ✉email: j.a.rising@lse.ac.uk

Extreme temperatures under climate change are predicted to reduce average yields for several of the United States' major crops[1–4]. However, these impacts can vary across space, with some areas showing benefits from increases in moderate temperatures and increased evapotranspiration under irrigation[5,6]. As climate shifts, these changes in productivity will drive farmers to change crops and move into new areas[7]. Understanding the extent of these regional changes in agricultural productivity and how they influence future cropping decisions is a central question for the impacts of climate change on agriculture[8,9]. Crop shifting may be able to attenuate climate impacts, but the potential benefits depend on the distribution of impacts, the total availability of productive land, and the costs of switching crops.

In this paper, we explore the potential redistribution of six crops in the United States as an adaptation to climate change. We approach the crop shifting problem as a spatial optimization problem to maximize profits, following Polasky et al.[10] and Devineni and Perveen[11]. Our key innovation consists of providing a new empirical approach which better supports this form of crop shifting analysis, by providing estimates of the potential for crops as they move into new areas.

Empirical agricultural crop models use variation in the weather to explain yearly variation in crop yields[5,12,13]. Local agricultural management decisions are detailed and dynamic in a way that is unavailable to scientists working at large spatial scales. Econometric techniques allow these unobserved differences between regions to be accounted for with local baselines. However, these techniques have two consequences that undermine their ability to model the crop shifting process. First, they can model changes in yields, but not yield levels, since this information is factored out with region-specific baselines. As a result, crop productivity in regions that are not observed growing the crop cannot be determined. Second, they have a resolution-variance trade-off, whereby interaction terms that allow the relationship between weather and yield to vary by region necessarily reduce the precision of the estimated relationship within each region and may lead to over-fitting.

In this paper, we develop a Bayesian approach which addresses both of these challenges. As with econometric models, yields are predicted with a log-linear model, with terms for the non-linear effect of temperatures, crop water deficits, and a linear technology trend. In our model, the parameters of the model are allowed to vary for each high-resolution region, represented here with US counties. To constrain this regional variation in parameters and predict parameters in new regions, the expected values of each region's coefficients and of the regional intercept are modeled as a linear combination of a set of spatial covariates in a hierarchical Bayesian model[14,15]. The method allows "partial pooling", whereby the degree to which regions are pooled to estimate a single national set of parameters is determined by the data: if the data support idiosyncratic regional differences in temperature sensitivity, for example, very little pooling between regions will be used and the parameters for each region will be estimated separately. The covariates used to predict variation in the sensitivity to weather are the annual mean temperature, isothermality (diurnal range divided by annual temperature range), temperature seasonality (standard deviation over months), annual precipitation, precipitation seasonality (coefficient of variation across months), and irrigation fraction by crop. Both the region-specific weather coefficients and the model of how those coefficients vary over space are estimated simultaneously. In comparison to a least-squares regression approach, the hierarchical Bayesian approach is more efficient than a two-stage estimation process and allows more regional variation than a regression model with interacted coefficients. Using the resulting model, we forecast yield losses for

all six crops studied, when applied to current cropping patterns. We use the modeled yields for crops outside of their historical growing regions to estimate the potential for crop switching to mitigate these losses. In aggregate, agricultural losses for the crops we study can be reduced by half, but some regions become unsuitable for any of the crops.

## Results

**Spatial variation in climate sensitivity**. We fit the Bayesian yield model to yield observations for United States counties from 1949 to 2009 for six crops: barley, corn, cotton, soybeans, rice, and wheat. The covariate model is used to predict weather response functions and yields in new locations for each crop. The coefficients for extreme degree-days, a key driver behind climate impacts, are shown in Fig. 1 (others are in Supplementary Figs. 10–15).

The spatial patterns for the effects of extreme temperatures vary by crop. Corn and cotton show less sensitivity to extreme temperatures in the southern US, reflecting adaptation in seed varieties and farming practices to minimize losses. For wheat and barley, adaptation is dependent upon water availability, with higher sensitivity in dry regions. We find that a fairly low degree of partial pooling was applied so that the estimated parameters for the county-specific models vary considerably. The 95% range of the estimated coefficients on extreme temperatures is 2 (rice) to 12 (cotton) times the standard error of the average coefficient. Much of the variation in coefficients is explained by county mean temperature, suggesting existing adaptation to higher temperatures. The portion of the variation in crop yield sensitivity to extreme temperatures that is explained by mean temperature varies from 8% for soybeans to 63% for cotton. Finally, coefficients vary slowly across space, showing spatial correlations up to 2000 km (Supplementary Note 7).

**Comparison of crop modeling approaches**. To validate the crop models, we compare the coefficients of determination (unadjusted $R^2$) for each crop to the results of a series of panel econometric regressions, mapping out the range between the model used in Schlenker and Roberts[5] and a regression-based equivalent to our analysis using covariate interactions. Since we are interested in the ability of the model to predict future years, we also perform cross-validation, by fitting the model to data from 1949 to 1994 and evaluating it on yields from 1995–2009. These results are shown in Table 1.

Applied to data from all years, the Bayesian model performs similarly to the most flexible ordinary least-squares (OLS) models with linearly varying coefficients. However, these same OLS models are prone to over-fitting and show large decreases in their $R^2$ under cross-validation. OLS models with constant coefficients across all counties perform better under cross-validation. While the Bayesian models also show reduced predictive capacity under cross-validation, they out-perform all OLS models for three of the crops. In all cases, they have a greater $R^2$ than similarly flexible OLS models. This is due to the idiosyncratic differences between coefficients in different counties that are permitted in the Bayesian model.

**Shifting cultivation under climate change**. Next, we use the Bayesian model to identify the optimal cultivation patterns now and in the future. We use the yield model with constant error variance (Table 1, column 6) to limit the variance in unobserved counties. Since cultivation costs and prices vary across the United States, we use profit (local price times predicted yield, minus management costs) in USD acre$^{-1}$ to determine the best crop. Costs and prices are from USDA Economic Research Service[16] for 2010, adjusted when necessary to make the locally optimal crop

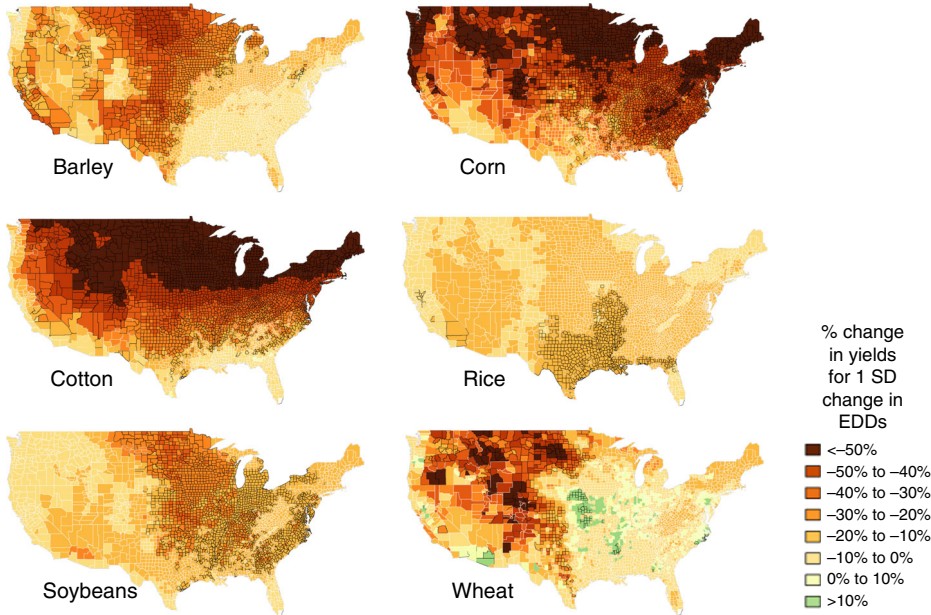

**Fig. 1 The effect of extreme degree-days on yields, across counties, and crops.** The displayed coefficients are for the effect of a 1 standard deviation change in extreme degree-days (EDDs) on log yield, interpretable as the fractional effect on yields. The response to extreme temperatures is predicted even in areas where the crop is not currently grown. Each crop has a different growing season and extreme degree-day cut-off, so that the model coefficient is normalized by a different standard deviation per crop (240 EDDs/SD for barley, 65 for corn, 40 for cotton, 63 for rice, 64 for soybeans, and 82 for wheat). County outline color indicates the confidence level (solid black outline: >95% of posterior draws have the same sign; thin white outline: <67% of posterior draws have the same sign).

**Table 1 Comparison of the predictive power of OLS and Bayesian yield models.**

**Model specifications**

|  | OLS 1 | OLS 2 | OLS 3 | OLS 4 | Bayes 1 | Bayes 2 |
|---|---|---|---|---|---|---|
| Intercepts | Uniform | County | Interacted | County | Partial | Partial |
| Coefficients | Uniform | Uniform | Interacted | Interacted | Partial | Partial |
| Error variance | Uniform | Uniform | Uniform | Uniform | County | Uniform |

**$R^2$ by model: estimated and evaluated on all years**

|  | OLS 1 | OLS 2 | OLS 3 | OLS 4 | Bayes 1 | Bayes 2 |
|---|---|---|---|---|---|---|
| Barley | 0.36 | 0.71 | 0.57 | 0.75 | 0.74 | 0.75 |
| Corn | 0.48 | 0.76 | 0.65 | 0.78 | 0.81 | 0.82 |
| Cotton | 0.32 | 0.64 | 0.55 | 0.70 | 0.68 | 0.69 |
| Rice | 0.75 | 0.84 | 0.81 | 0.84 | 0.85 | 0.85 |
| Soybeans | 0.47 | 0.72 | 0.65 | 0.76 | 0.78 | 0.79 |
| Wheat | 0.42 | 0.71 | 0.56 | 0.73 | 0.76 | 0.76 |

**$R^2$ by model: estimated on 1949–1994, evaluated on 1995–2009**

|  | OLS 1 | OLS 2 | OLS 3 | OLS 4 | Bayes 1 | Bayes 2 |
|---|---|---|---|---|---|---|
| Barley | −0.11 | 0.43 | 0.20 | 0.45 | 0.48 | 0.46 |
| Corn | −0.09 | 0.20 | 0.07 | −1.05 | 0.27 | 0.17 |
| Cotton | 0.07 | 0.31 | 0.14 | −37.50 | 0.21 | 0.12 |
| Rice | 0.20 | 0.37 | 0.12 | −1.59 | 0.19 | 0.14 |
| Soybeans | 0.26 | 0.47 | 0.39 | −16.27 | 0.53 | 0.48 |
| Wheat | 0.16 | 0.49 | 0.31 | 0.47 | 0.51 | 0.50 |

Table cells show $R^2$ by crop and model specification, using all data (top) and under cross-validation on 1995–2009 (bottom). The first four columns are ordinary least-squares (OLS) specifications, variously including region-specific intercepts and covariate interactions. The last two columns are for the Bayesian model, with partially pooled intercepts and coefficients, either allowing each county to have a different variance (Bayes 1) or constraining all to have the same variance (Bayes 2). In all cases, $R^2 = 1 - \frac{\sum (y_i - \hat{y}_i)^2}{\sum (y_i - \bar{y}_i)^2}$, where $y_i$ is the observed log yield for county-year $i$. $\hat{y}_i$ is the point estimate for OLS and the posterior prediction for the mean MCMC parameter draw for the Bayesian model, and $\bar{y}_i$ is the average across all observations of $y_i$.

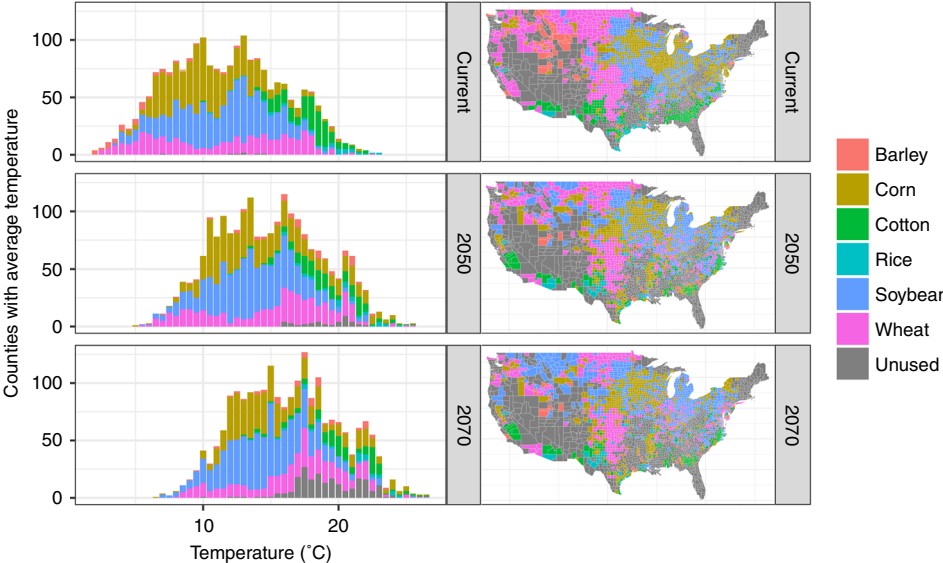

**Fig. 2 Optimized crop patterns for each period across temperature and space.** Periods are shown on different rows, and the distribution across temperature is shown on the left and across space on the right. When plotted as a distribution across temperature, the climatic annual mean temperature of each county is used, and the distribution is across counties. Unused (gray) regions are displayed where none of the six crops are planted at baseline or where total profits are maximized by leaving land fallow later in the century.

according to profits match the most widely planted observed crop. Since we do not account for alternative uses of land, we constrain the crops to only be cultivated in the future in areas currently used for at least one of the six crops. Changes in future crop production can also result in general equilibrium effects on prices[9]. Here, we avoid significant price changes by limiting the total land used by each crop to not exceed current nation-wide totals.

Applied to the current climate, crops are grown in characteristic temperature ranges, as shown in Fig. 2 (top). Barley and wheat are mainly grown in cooler counties, while cotton is grown in the warmest areas. However, these suitability envelopes are not exclusive, with some barley and (winter) wheat grown at higher temperatures. Although the optimization is calibrated to prefer the crop currently most planted in each county, 16% [14–18%] (ranges in brackets display the 95% credible interval throughout) of counties do experience changes under the optimization, as secondary crops are replaced with the optimal crop, and then these secondary crops are shifted to other counties. This results in a 13% [8–37%] increase in total profits (Fig. 3 and Supplementary Fig. 29). The largest changes result from swaps between soybean and corn, which are commonly grown in rotation (excluding corn–soy swaps, 5% [4–6%] of counties show changes). We then use a suite of CMIP5 models to project these changes in optimal crops forward under RCP 8.5 and report outcomes in 2050 and 2070 including both climate and statistical uncertainty (Fig. 2 and Supplementary Note 16 and Supplementary Tables 13–15). Corn retains its enormous area (by construction, so long as corn profits are positive), but becomes less concentrated in the Midwest. Soybeans show a gradual movement north, replacing spring wheat and barley. The wheat lands of the Great Plains see a gradual hollowing-out, while winter wheat moves up from the south along the Mississippi. Cotton is grown at higher latitudes, becoming the dominant crop in southern California. At the same time, lands in the southern US that are not profitable for any crop expand. These tend to be at the higher end of the temperature distribution, and account for 5% of the included land area by 2070. We do not observe a uniform movement to higher latitudes, because of regional variation in climate and the constraint against crops moving into new areas (Supplementary Note 17 and Supplementary Figs. 30 and 31).

**Economic outcomes of adaptation**. Figure 3 (top) shows the amount of switching between crops to maximize profits. Large portions of corn and soybean cultivation continue to swap in 2050, but changes from 2050 to 2070 are more minor. By 2070, 53% [39–67%] of counties experience crop switching (36% [21–51%] excluding corn–soy swaps).

A comparison of the effects of optimization on profits is shown in Fig. 3 (bottom). In the absence of optimization, total estimated profits fall from $45.7 [$44–52] billion to $35.8 [$24–50] billion in 2050 and $31.4 [$19–48] billion in 2070, a 31% decrease [59↓–5%↑]. With optimization, profits in 2010 were predicted to be able to increase to $51.8 [$49–63] billion. However, they fall below current profits by 2050 and by 2070, even with further optimization, they fall to $38.6 [$28–54] billion, still 16% below [38↓–18%↑] observed levels. Relative to the profits of optimally reallocated crops in the current period, percentage losses from climate change are greater, 26% below [45↓–4%↑] the peak.

Behind these profits are both increases and decreases in individual crop production. Production is predicted to be able to increase for most crops under current conditions and optimal planting, ranging from small decreases for soy (2% [4–1%]) to large production increases for barley (26% [11–44%]). By 2070, however, decreases in total production are shown for barley (9% [22↓–4%↑]), corn (37% [74↓–10%↑]), rice (2% [30↓–37%↑]), and soybeans (6% [16↓–5%↑]) relative to observed production. These are offset by increases from cotton (73% [20–192%]) and wheat (2% [26↓–28%↑]). These results do not extrapolate the historical trend in crop yields into the future, to isolate the relative role of climate change (we explore this in Supplementary Notes 18 and 19, Supplementary Tables 16–18, and Supplementary Figs. 31–34).

In the default model, we assume that there are no additional barriers or frictions involved in switching crops, and explore the effects of imposing a range of crop switching costs in Supplementary Note 20 and Supplementary Fig. 35. Switching costs of $180/acre reduce reallocation changes by half, against average cultivation costs between $123/acre (barley) and $499/ acre (rice). As switching costs increase, optimal losses converge to the losses without crop reallocation. Since optimal profits in 2050 are below current profits, losses will persist under any level of switching costs.

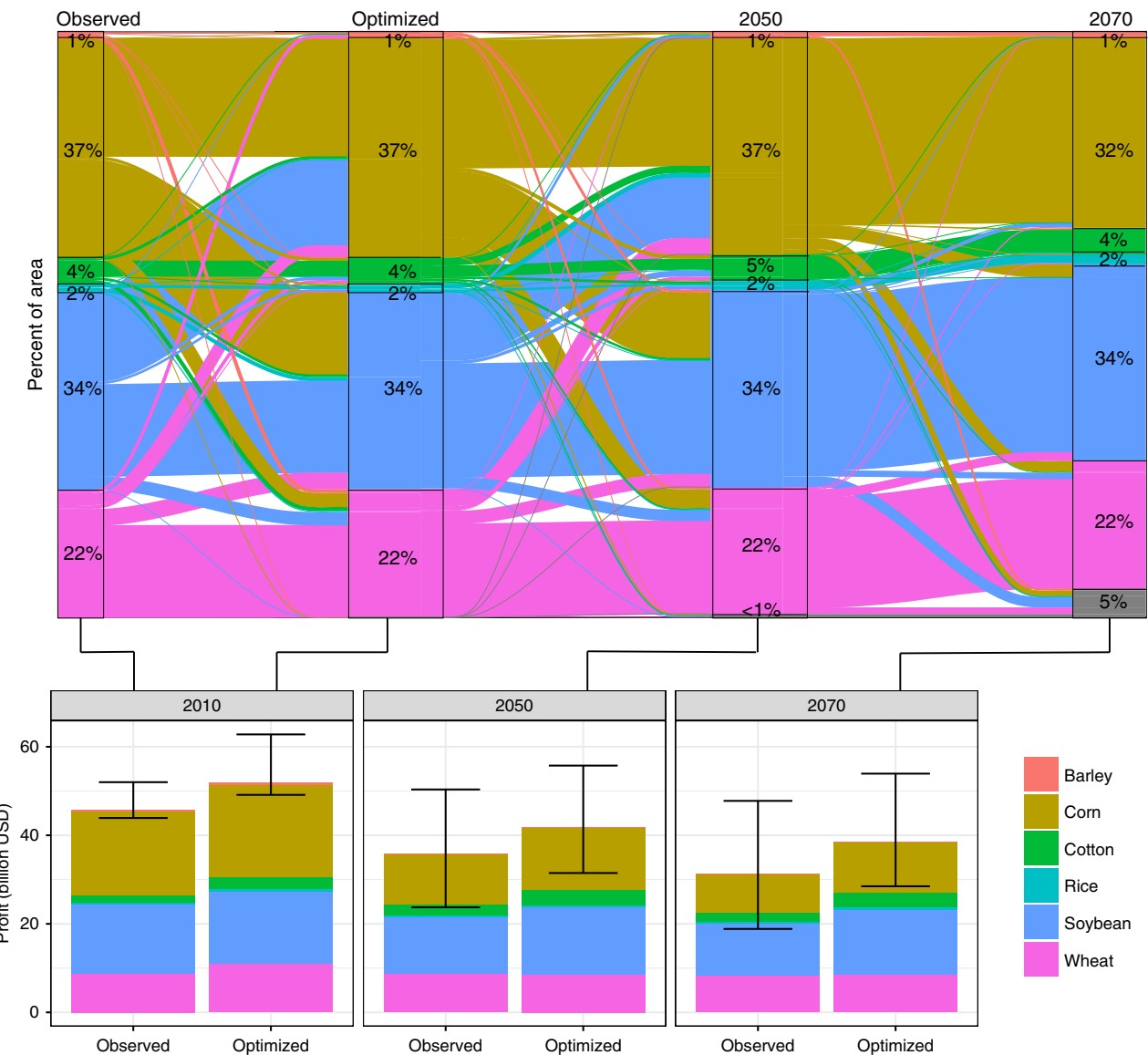

**Fig. 3 Adaptation outcomes accounting for crop shifting.** (Top) The portion of the area allocated to each crop, under the optimization, in percent labeled boxes. Flows between the allocations show the portion of the area previously allocated to the crop on the left, and flowing into its new allocation on the right. The difference between observed and optimized crop allocations (first transition) is due to replacing secondary crops with primary crops. (Bottom) Profits under observed and optimized crop allocations for the current climate (first box), 2050, and 2070. The first bar in each column gives estimates of profits without relocation of crops, and the second bar is with optimization. The bars show mean profits across posterior draw Monte Carlo optimizations and the error ranges show 95% credible intervals.

## Discussion

Agriculture is one of the most exposed sectors to the impacts of climate change, and adaptation through irrigation investments, agricultural research, and new management practices can require decades of planning. A better understanding of the potential for adaptation is needed for farmers and policy-makers to make long-term decisions. We show that adaptation through the movement of crops can reduce climate change losses, but it does not eliminate them.

We have focused here on the expected losses to agricultural production, but several other dimensions of impacts are embedded in these numbers. Nation-wide average decreases in yields are likely to emerge through more years of unforeseen crop failures and through regional devastation. The crop switching actions projected in this paper would cause disruptions to farmers, food supplies, and environmental habitats. Even if crops are mobile,

farmers may not be. In particular, farmers who work on the 5% of cultivated land that becomes economically untenable under our model will need to identify new crops or land uses outside the scope of this study.

Our empirical model only captures adaptation practices currently employed to respond to within-year shocks of high temperatures. Future work is needed to explicitly account for the potential and limits of irrigation expansion, long-term investment in adaptation, and to distinguish the benefits of $CO_2$ fertilization from the long-term trend. While we consider multiple sources of uncertainty in the outcomes, we do not account for risk aversion, unexpected weather shocks, or the multi-year consequences of crop failures.

Our optimization approach assumes perfect knowledge of crop weather responses and that observed weather will correspond to the expected climate. As such, our results should be considered a

frontier of possibility, assuming that crop yields respond to temperatures in the future as they have in the past. The cropping patterns shown in our current and future results should not be taken as recommendations, since many details at the field and farmer level are not included.

Our results show considerable potential from crop switching to avoid some of the damages from climate change. These opportunities are driven both by differences in how temperatures may change in different regions as well as differences in the sensitivity of crops to higher temperatures. However, the remaining losses imply that crop switching is not a panacea and that new seed varieties and new adaptation practices are needed to support farmers and meet the food demands of the future.

## Methods

**Climate and crop data**. County-level yield data[17] and weather data[18] cover the contiguous US for 1949–2009. Annual crop water deficit indices are calculated as in ref. [19], and growing degree-days and extreme degree-days are calculated as in ref. [5]. County-level constant covariates used in the multilevel model consist of annual mean temperature, isothermality (diurnal range divided by annual temperature range), temperature seasonality (standard deviation over months), annual precipitation, precipitation seasonality (coefficient of variation across months), all from ref. [20], and irrigation fraction by crop, from ref. [21]. Additional details are in Supplementary Tables 1 and 2 and Supplementary Notes 1 and 2.

**Multilevel Bayesian crop model**. We fit a Bayesian model which represents log-yields as a linear model of crop water deficits, growing and extreme degree-days, and a linear trend. The coefficients of this model are allowed to vary by county, with an expected value for each county-specific coefficient equal to a linear model of the six county-level constant covariates listed above. That is, for each crop,

$$\log(Y_{it}) \sim N(\alpha_i + \beta_i^1 t + \beta_i^2 \mathrm{CDI}_{it} + \beta_i^3 \mathrm{GDD}_{it} + \beta_i^4 \mathrm{EDD}_{it}, \sigma_i^2)$$

$$\alpha_i \sim N\left(a_0 + \sum_{j=1}^{\#\,\mathrm{covar}} b_j^0 \mathrm{covar}_{ij}, \sigma_\alpha^2\right)$$

$$\beta_i^k \sim N\left(a_k + \sum_{j=1}^{\#\,\mathrm{covar}} b_j^k \mathrm{covar}_{ij}, \sigma_{\beta^k}^2\right)$$

where $Y_{it}$ is the yield in county $i$ in year $t$, $\mathrm{CDI}_{it}$ is the water deficit predictor, $\mathrm{GDD}_{it}$ is growing degree-days, $\mathrm{EDD}_{it}$ is extreme degree-days, and $\mathrm{covar}_{ij}$ is the value of covariate $j$ for county $i$. All other parameters are fit in the model. Additional details are in Supplementary Note 3 and model validation is shown in Supplementary Notes 4 and 5, Supplementary Tables 3 and 4, and Supplementary Figs. 1–9. Additional fitted results are shown in Supplementary Notes 6 and 7, Supplementary Tables 5–7, and Supplementary Figs. 16–20.

**Comparison to OLS models**. The predictive power of the Bayesian model is compared to multiple least-squares (OLS) regressions. The regression terms are combinations from the following intercepts and coefficients columns, according to the table header in Table 1:

$$\log(Y_{it}) = \begin{pmatrix} \underline{\mathrm{Intercepts:\ one\ of}} \\ \underline{\mathrm{Uniform}} \\ \alpha \\ \underline{\mathrm{County}} \\ \alpha_i \\ \underline{\mathrm{Interacted}} \\ \alpha_0 + \sum_{j=1}^{\#\,\mathrm{covar}} \alpha_j \mathrm{covar}_{ij} \end{pmatrix} + \begin{pmatrix} \underline{\mathrm{Coefficients:\ one\ of}} \\ \underline{\mathrm{Uniform}} \\ \beta_0^1 t + \beta_0^2 \mathrm{CDI}_{it} + \beta_0^3 \mathrm{GDD}_{it} + \beta_0^4 \mathrm{EDD}_{it} \\ \underline{\mathrm{Interacted}} \\ \beta_j^1 t + \beta_0^2 \mathrm{CDI}_{it} + \beta_0^3 \mathrm{GDD}_{it} + \beta_0^4 \mathrm{EDD}_{it} \\ + \sum_{j=1}^{\#\,\mathrm{covar}} \beta_j^1 \mathrm{covar}_{ij} t + \beta_j^2 \mathrm{covar}_{ij} \mathrm{CDI}_{it} \\ + \beta_j^3 \mathrm{covar}_{ij} \mathrm{GDD}_{it} + \beta_j^4 \mathrm{covar}_{ij} \mathrm{EDD}_{it} \end{pmatrix} + \epsilon_{it}.$$

Under cross-validation, both the Bayesian and OLS models are fit only to data prior to 1995, and the $R^2$ value is computed only on data from 1995 to 2009. Additional details are in Supplementary Note 8 and Supplementary Tables 8 and 9. Extensions to the model are described in Supplementary Notes 9 and 10, Supplementary Tables 10 and 11, and Supplementary Figs. 21–24.

**Land-use optimization model**. Optimized land use is projected using a linear programming model, which determines the profit-maximizing distribution of crops under the yields estimated by the Bayesian model. The optimization problem is,

$$\max_{\{A_{ict}\}} \sum_c \sum_i (p_{ic} \hat{Y}_{ict} - o_{ic}) A_{ict},$$

for the area of crop $c$ in county $i$ and period $t$ given by $A_{ict}$ and the price $p_{ic}$ and cultivation costs $o_{ic}$ are drawn from ref. [16] for 2010. The optimization is performed separately for each draw from the posterior estimate of yield, $\hat{Y}_{ict}$. Yields are adjusted to account for the irrigation capacity of the destination county. The

optimization is constrained such that,

$$\sum_c A_{ict} \le \sum_c \bar{A}_{ic} \quad \forall i \quad \text{No additional land is appropriated to farming in any county,}$$

$$\sum_i A_{ict} \le \sum_i \bar{A}_{ic} \quad \forall c \quad \text{No additional land is appropriated to any crop,}$$

where $\bar{A}_{ic}$ is the area used by crop $c$ in county $i$ in 2010.

When the optimization is applied to observed yields and reported cultivation costs, 40% of counties are assigned crops that do not match observed planting. We treat this as reflecting hidden costs and adjust the cost values for these counties to make the observed crops optimal.

Future weather data for 2050 and 2070 is calculated using downscaled and bias-corrected CMIP5 results those years from ref. [22] for the 17 GCMs included in ref. [20]. Additional details are in Supplementary Notes 11–15, Supplementary Table 12, and Supplementary Figs. 25–28.

**Reporting summary**. Further information on research design is available in the Nature Research Reporting Summary linked to this article.

## Data availability
The data used in making the charts and tables in this paper are available at https://doi.org/10.5281/zenodo.3889144.

## Code availability
All models and display codes are available at https://doi.org/10.5281/zenodo.3909637. The optimization model is constructed using the land-use component of the open-source AWASH 2.0 water-energy-food model, available at https://github.com/AmericasWater/awash. The Bayesian model uses JAGS 4.3.0 and the optimization model uses Gurobi Optimizer 9.0.2.

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

## Acknowledgements

This research was supported by NSF grant 1360446 (Water Sustainability and Climate, Category 3). Rising acknowledges the support of the Grantham Research Institute on Climate Change and the Environment at the London School of Economics, and the ESRC Centre for Climate Change Economics and Policy (CCCEP) (ref. ES/R009708/1). The statements contained within the manuscript/research article are not the opinions of the funding agency or the U.S. government but reflect the authors' opinions. We thank Mr. Paul Alabi of the City College of New York (CCNY) for ensuring uninterrupted operation of the high-performance computing facilities at CCNY that facilitated the final Bayesian computations during the COVID-19 pandemic lockdown in New York City.

## Author contributions

All authors designed and performed the analysis and wrote the text. N.D. performed the Bayesian modeling and J.R. performed the optimizations.

## Competing interests

The authors declare no competing interests.
