## [Peer Review File · Nature Communications]

Reviewer's Comments

Reviewer #1

James Rising and Naresh Devineni describe the effects of crop switching as a tactic for reducing impacts of climate change on agricultural production in the United States. Their manuscript is well prepared. The topic is broadly important and would engage the readership of *Nature Communications*.

I found many flaws with the Bayesian analysis. These shortcomings are as follows.

1. The authors claim to “develop” a Bayesian model. This is a stretch. Actually, they apply a model that is taught in elementary Bayesian texts (Gelman and Hill, 2009).
2. No report of a Bayesian analysis is complete without an complete expression for the posterior and fully factored joint distribution, which fails to appear here. I routinely return Bayesian papers without review, in my own editorial work, if they lack this critical component. It is impossible to fully appreciate what was done in its absence.
3. Parameters of prior distributions must be reported and justified.
4. The fundamental assumption of all model-based inference is that the model is capable of giving rise to the data. There are a variety of methods of checking that this assumption is met, notable among them, posterior predictive simulation. (Gelman and Hill, 2009; Hobbs and Hooten, 2015; Conn et al., 2018). The model reported was not checked.
5. Markov chain Monte Carlo approximations of posterior distributions can fail to converge. This is especially problematic for highly parameterized models like this one. Inference is not reliable in the absence of convergence. There are a variety of tests that can be used to assure convergence occurred. None appear in this paper.
6. Spatially structured observations like those reported here can lead to spatial autocorrelation in residuals. Reliable analysis must model this spatial structure if it exists. Failing to do so will lead to inference that is excessively optimistic (Cressie and Wikle, 2011). There is no evaluation of spatial structure in model residuals.
7. The paper is peppered with point estimates of % change. There is no reason that these cannot be accompanied by Bayesian credible intervals, preferably highest posterior density intervals.

A great strength of Bayesian analysis using MCMC is the ability to make such inference on derived quantities. The authors should exploit that strength. (Hobbs and Hooten, 2015).

8. There is no model for uncertainty in the predictor variables, which means that the authors are forced to assume that they are measured without error. This heroic assumption can lead to conclusions that are fundamentally false (Clark, 2007).
9. R^2 is a notoriously unreliable statistic for model comparison. See (Hooten and Hobbs, 2015) for proper approaches to model selection in the Bayesian framework.

Literature Cited

- Clark, J. M., 2007. Models for ecological data. Princeton University Press., Princeton, New Jersey, USA.
- Conn, P. B., D. S. Johnson, P. J. Williams, S. R. Melin, and M. B. Hooten, 2018. A guide to Bayesian model checking for ecologists. *Ecological Monographs* **88**:526–542.
- Cressie, N. and C. K. Wikle, 2011. Statistics for spatio-temporal data. Wiley.
- Gelman, A. and J. Hill, 2009. Data analysis using regression and multilevel / hierarchical modeling. Cambridge University Press, Cambridge, UK.
- Hobbs, N. T. and M. B. Hooten, 2015. Bayesian models: A statistical primer for ecologists. Princeton University Press, Princeton New Jersey, USA.
- Hooten, M. B. and N. T. Hobbs, 2015. A guide to Bayesian model selection for ecologists. *Ecological Monographs* **85**:3–28.

Reviewer #2 (Remarks to the Author):

“Crop switching reduces agricultural losses from climate change in the United States by half” estimates a model of yields and then explores what crop choices that model would predict today and into the future as climate changes across farmers in the United States. The paper predicts that crop choice would encourage crops to migrate and increase yields in response to climate change. The authors estimate that crop switching would cut future yield losses from climate change in half.

I am very sympathetic with the author’s conclusions but I am not convinced that the authors have developed a reliable methodology.

First, they assume that temperature has a linear effect on yields whereas the literature has clearly

shown the effect is nonlinear. Even their results suggest that their model would improve if it included a quadratic term on GDD (they find that the coefficient on GDD falls with higher temperature). They should carefully examine when their county coefficients change with the level of the variable.

Second, the text claims that crops would move to higher latitudes in response to warming but only one of their six crops, soybeans, moves to higher latitudes (and also west). Wheat and corn move south according to Figure 2. Cotton abandons its southern position to move to the Far West. What explains these changes?

Third, it appears that the model is not that good at predicting where crops are now. As best as I can tell from the paper, the model predicts the correct crops today about three fourths of the time when it is estimated on the actual outcomes. But when the model is estimated using one period of data, and then tested on a subsequent period, it chooses the correct crop less than half the time.

Fourth, the model of wheat suggests that wheat thrives on extreme temperatures but not growing degree days. Given that the range of temperatures for each crop were chosen so that degree days are beneficial and extreme temperatures are harmful, this is nonsensical. It is possible that they are confusing winter wheat and spring wheat which are grown over different months.

Fifth, it is very difficult to determine what their methodology actually is. The description of their approach has typos and vague sentences that make it unclear, in sharp contrast to their description of alternative econometric models. It appears that they are estimating a model with interactions in two stages. Why did they adopt a two stage model if their parameters can be estimated more efficiently simultaneously?

The maps of coefficients (Figures 1-6) appear to be showing that the differences in coefficients across space is revealing interaction terms across variables. It is not clear why the model does not adopt the simpler econometric alternative as shown in the appendix and simply include interaction terms.

Sixth, the simulation modeling seems to depend on arbitrary assumptions such as no technical change, fixed irrigation fractions, fixed local prices, fixed costs per crop, and fixed aggregate acreages (or aggregate yields?) of crops. They also appear to alter the observed cost of growing each crop in each county in order to force the model to match observed crop choice. It is not clear why the reader should have any confidence in these forecasts.

Seventh, their description of crop switching cost is hard to follow. How do they measure the cost of switching? Or do they actually calculate the benefit of switching so that these "switching costs" are actually the benefit to any farmer who does switch?

Eighth, the text does not do a good job of describing the results. For example, they do not have a discussion of how crops move in Figure 2 although that is the purpose of the paper. How should one interpret Figure 1 if the crop is not grown in most counties in the map? What does the top Figure 3 say if the forecast is constrained to keep aggregate acreage (or aggregate yields?) of each crop the same? Does the comparison of actual versus optimal profit in 2010 in bottom of Figure 3 imply that the authors know how to manage every cropland acre better than the farmers that live there? In contrast, is the comparison in future years show the difference between assuming farmers do not change crop choice versus they do?

They claim but do not show that "extreme degree days [are] the key driver behind climate impacts". Does EDD explain more than GDD and CID?

What is Figure 7 in the Appendix telling the reader?

Robert Mendelsohn

Reviewer comments and responses

Reviewer 1:

Thank you for your helpful comments and close reading of the Bayesian modeling approach. We have provided more information and extended how we report results in the main paper, as detailed below.

The authors claim to “develop” a Bayesian model. This is a stretch. Actually, they apply a model that is taught in elementary Bayesian texts (Gelman and Hill, 2009). Gelman, A. and J. Hill, 2009. Data analysis using regression and multilevel / hierarchical modeling. Cambridge University Press, Cambridge, UK.

We agree. In the revised manuscript, we have adjusted our wording to avoid suggested that we in any way developed the hierarchical Bayesian modeling approach. We do consider the development of the full approach, consisting of applying a Bayesian hierarchical model as we have and pairing it with economic data and an optimization model, to be one of the key contributions of the paper. However, we have changed text in several places to clarify this, including adjusting the abstract text to read “We develop an approach to estimate the economic potential of crop reallocation using a Bayesian hierarchical model of yields.”

No report of a Bayesian analysis is complete without an complete expression for the posterior and fully factored joint distribution, which fails to appear here. I routinely return Bayesian papers without review, in my own editorial work, if they lack this critical component. It is impossible to fully appreciate what was done in its absence.

Thank you for pointing this out. We acknowledge the importance of having these details in the manuscript. In the methods section of the revised manuscript, we now added the full likelihood expression in equation (6).

Parameters of prior distributions must be reported and justified.

We now report the prior distributions in equation (5). Essentially, for the variance terms we assumed a wide uniform prior and for the coefficients at the second level we assumed non-informative normal priors.

The fundamental assumption of all model-based inference is that the model is capable of giving rise to the data. There are a variety of methods of checking that this assumption is met, notable among them, posterior predictive simulation. (Gelman and Hill, 2009; Hobbs and Hooten, 2015; Conn et al., 2018). The model reported was not checked. Hobbs, N. T. and M. B. Hooten, 2015. Bayesian models:

A statistical primer for ecologists. Princeton University Press, Princeton New Jersey, USA. Conn, P. B., D. S. Johnson, P. J. Williams, S. R. Melin, and M. B. Hooten, 2018. A guide to Bayesian model checking for ecologists. *Ecological Monographs* 88:526–542.

Thank you for mentioning this. The posterior predictive checks are now presented in section 2 of the methods part. For each of the six crops, we randomly selected two counties from the regions where they are currently cultivated and compared the posterior predictions with the observed crop yields. We show this using a series of box plot diagrams depicting the posterior distributions of yields from 1949 - 2009 from the cross-validated model that uses 1949 - 1994 as the training set to predict yields for 1995-2009.

Markov chain Monte Carlo approximations of posterior distributions can fail to converge. This is especially problematic for highly parameterized modes like this one. Inference is not reliable in the absence of convergence. There are a variety of tests that can be use to assure convergence occurred. None appear in this paper.

This is an important point. We did check for converge of the parameters. However, for the sake of brevity, in the original manuscript, we did not present the convergence statistics. Now, in the revised manuscript (section 2.2 of the methods), we present a summary table of R-hat for all the models. The median R-hat value across all parameters and across all the counties is shown in the table.

Spatially structured observations like those reported here can lead to spatial autocorrelation in residuals. Reliable analysis must model this spatial structure if it exists. Failing to do so will lead to inference that is excessively optimistic (Cressie and Wikle, 2011). There is no evaluation of spatial structure in mode residuals. Conn, P. B., D. S. Johnson, P. J. Williams, S. R. Melin, and M. B. Hooten, 2018. A guide to Bayesian model checking for ecologists. *Ecological Monographs* 88:526–542.

It is true that we did not explicitly model the spatial auto-correlation in the residuals. However, we do capture some drivers of spatial correlation through the level 2 model where we model the coefficients (response to climate) using geographic covariates. This provides a modest replication of the spatial correlations in the yields. To identify issues coming from ignoring autocorrelation in the residuals, we perform an analysis that looks at the differences between observed and predicted spatial correlation. We present these results in Figure 2 of the Methods section. We do acknowledge that having an explicit model to estimate spatial correlation is important. For the objectives of this work, we considered that our multi-level approach should suffice.

The paper is peppered with point estimates of % change. There is no reason that these cannot be accompanied by Bayesian credible intervals, preferably highest posterior density intervals. A great strength of Bayesian analysis using MCMC is the ability to make such inference on derived quantities. The authors should exploit that strength. (Hobbs and Hooten, 2015).

This is an important point, but requires some nuance in our case. Almost all of the percent changes we report in the main paper are from the optimization exercise, for which we collapse uncertainty across both the MCMC draws and GCMs. There are two types of results that we do extract directly from the Bayesian model. (1) We estimate weather response coefficients by county. The uncertainty in these, which is estimated as quantiles over the MCMC draws, is shown in the maps in SI figures 3 – 8, showing uncertainty using different county border lines. (2) The hyper-parameters were not previously reported in our paper, but we have added these as SI table 2. In this table, we report 95% equal-tailed credible intervals.

However, we have the opportunity to use MCMC draws to also generate uncertainty intervals for the results of the optimization modeling, and we now do this throughout those results in the paper. Specifically, rather than using the mean predicted yields from the Bayesian model, we treat every MCMC draw as a unique optimization problem. Our average values and credible intervals are reported across these final metrics.

There is no model for uncertainty in the predictor variables, which means that the authors are forced to assume that they are measured without error. This heroic assumption can lead to conclusions that are fundamentally false (Clark, 2007). Conn, P. B., D. S. Johnson, P. J. Williams, S. R. Melin, and M. B. Hooten, 2018. A guide to Bayesian model checking for ecologists. Ecological Monographs 88:526–542.

We certainly acknowledge that exploring uncertainty in predictor variables is important. It could make an interesting modeling exercise. Given the scale and objectives of our work, we restricted ourselves to only modeling dependent variables. This is consistent with the econometric literature that we are aiming to compare our model with.

R² is a notoriously unreliable statistic for model comparison. See (Hooten and Hobbs, 2015) for proper approaches to model selection in the Bayesian framework. Hooten, M. B. and N. T. Hobbs, 2015. A guide to Bayesian model selection for ecologists. Ecological Monographs 85:3–28.

This is a good point, and we have struggled with the best way to address it. We now report DIC values for the Bayesian models in section 2.2 of the Methods. However, we have kept the R² values in the main text. The main reason for this is that the other models we want to compare to are not Bayesian, and R² can be calculated only from the observed and predicted responses. We considered reporting a DIC-equivalent for each model, using the maximum likelihood of the OLS models, but we believe that the values are far less interpretable.

Reviewer 2:

Thank you for all of these comments. They have been very helpful in making the paper clearer, and we appreciate the time you have taken to offer this feedback.

First, they assume that temperature has a linear effect on yields whereas the literature has clearly shown the effect is nonlinear. Even their results suggest that their model would improve if it included a quadratic term on GDD (they find that the coefficient on GDD falls with higher temperature). They should carefully examine when their county coefficients change with the level of the variable.

Thank you for this comment. The nonlinear responses of yields to temperature are indeed a key ingredient to our analysis, and we are sorry that this was not clear in the original text. Our use of a combination of GDDs and EDDs (extreme degree-days) follows the seminal nonlinear approach of Schlenker & Roberts, and was the representation of nonlinearity that Wolfram Schlenker recommended to us in personal communication. We have also explored your suggestion of estimating the model including an additional quadratic effect of GDDs, and the improvement in R^2 is generally less than 1%, and less than 2% in all cases except for our OLS baseline model without fixed effects.

To avoid this confusion, we have adjusted the text so that rather than saying that the model includes “terms for the effect of moderate and extreme temperatures”, we explicitly say it has “terms for the non-linear effect of temperatures”.

Second, the text claims that crops would move to higher latitudes in response to warming but only one of their six crops, soybeans, moves to higher latitudes (and also west). Wheat and corn move south according to Figure 2. Cotton abandons its southern position to move to the Far West. What explains these changes?

Thank you, this comment has driven us to analyze our the spatial patterns and their drivers more fully. Although we would expect a broad movement northward, you are correct that we do not see it. We now explicitly say this in the text: “We do not observe an orderly movement to higher latitudes, because of our constraint against crops moving into new areas (see SI 11).” As crops compete for limited high-latitude area, the highest value crops are planted, and low-latitude areas show less movement, except as a result of land being left fallow.

In SI 11, we study the shadow price of the constraint that limits the land available in each county to its current level. The shadow price reflects the potential increase in profits from relaxing this constraint by 1 Ha, which is the value of land for these six crops. This shows a clear north-south gradient, where counties below 40 N show broad decreases in land value.

Third, it appears that the model is not that good at predicting where crops are now. As best as I can tell from the paper, the model predicts the correct crops today about three fourths of the time when it is estimated on the actual outcomes. But when the model is estimated using one period of data, and then tested on a subsequent period, it chooses the correct crop less than half the time.

Our paper should have been clearer on this point, since there are multiple performance metrics representing different models. The 75% and 50% accuracy rates that you mention are the capacity of the yield model to predict yield as a function of weather variation. We are now more clear in the text that these values reflect yield accuracy, not planting choices, and split out planting decisions to a separate section.

Our optimization model then uses these yield models to predict planting choice. The optimization correctly identifies the planting in each county 83.3% of the time (94.3% of the time if corn-soybean swaps are ignored). The remaining difference between observed and optimized crops occurs because the optimization prefers planting a single, most profitable crop for each county (rather than multiple crops as observed). We now explain in the text, “Although the optimization is calibrated to prefer the crop currently most planted in each county, 17% [14 - 18\%] of counties do experience changes under the optimization, as secondary crops are replaced with the optimal crop, and then these secondary crops are shifted to other counties.”

Fourth, the model of wheat suggests that wheat thrives on extreme temperatures but not growing degree days. Given that the range of temperatures for each crop were chosen so that degree days are beneficial and extreme temperatures are harmful, this is nonsensical. It is possible that they are confusing winter wheat and spring wheat which are grown over different months.

In the previously submitted version of the paper, we applied winter wheat planting and harvesting dates for both winter and spring wheat yields, which resulted in the anomalies you noticed. As a result of fixing this error, the effect of an extreme degree-day reduces log yields by -0.0004, which is more consistent with expectations. Although the effect of growing degree-days is also negative for wheat, the effect is smaller (-0.0002). This is at odds with the beneficial effect prescribed in DSSAT, but in line with our OLS models of wheat production.

Fifth, it is very difficult to determine what their methodology actually is. The description of their approach has typos and vague sentences that make it unclear, in sharp contrast to their description of alternative econometric models. It appears that they are estimating a model with interactions in two stages. Why did they adopt a two stage model if their parameters can be estimated more efficiently simultaneously?

We apologize for the confusion from our previous methodology section. We have made many edits to both the paper and the supplementary materials which we hope clarify our approach. In particular, we do estimate the two “stages” of our yield model simultaneously. We have now added the clarification, “Both the region-specific weather coefficients and the model of how those coefficients vary over space are estimated simultaneously. In comparison to a least-squares regression approach, the hierarchical Bayesian approach is more efficient than a two-stage estimation process and allows more regional variation than an regression model with interacted coefficients.”

The maps of coefficients (Figures 1-6) appear to be showing that the differences in coefficients across space is revealing interaction terms across variables. It is not clear why the model does not adopt the simpler econometric alternative as shown in the appendix and simply include interaction terms.

Thank you for this question. We agree that the differences in space reflect the variation in interacting variables. The main reasons why we use the hierarchical Bayesian approach, rather than a simpler econometric model with interaction terms, are (1) the Bayesian approach performs better and (2) we want to allow idiosyncratic intercepts (like fixed-effects) but also model the expected intercept in a single stage. As shown in table 1 in the main paper, econometric models with interaction terms are less predictive under cross-validation than the Bayesian model. We now explain in the paper that this is due to the county-specific variation between coefficients in different counties that are permitted in the Bayesian model.

Sixth, the simulation modeling seems to depend on arbitrary assumptions such as no technical change, fixed irrigation fractions, fixed local prices, fixed costs per crop, and fixed aggregate acreages (or aggregate yields?) of crops. They also appear to alter the observed cost of growing each crop in each county in order to force the model to match observed crop choice. It is not clear why the reader should have any confidence in these forecasts.

Thank you for your comment. We consider the modeling decisions to fix irrigation use, prices, and total acreage as boundary-condition assumptions, which we need because of the limited scope of our analysis. Because we do not consider all possible land uses, the economic equilibrium in prices, or competing uses for water and labor or capital, we need exogenous scenarios to produce a consistent analysis, and keeping these constant seems the most defensible and interpretable. We recognize that there exist other analysis approaches which may allow these to be relaxed. The remaining assumption, that gross technology remains constant, we believe is useful for comparability. However, we also consider a continued evolution of technology along its historical path in SI 13.

It is true that we need to adjust costs to match observed planting. In response to your question, we have added a new table to explain the challenges of matching observed planting, SI table 7. Using costs and prices from the USDA Economic Research Service (ERS), without any modeling on our part, we find that 40% of counties are not planting economically optimal crops (that is, another of our six studied crops would provide greater profits, according to ERS). We take this to mean that there are dimensions of the costs that are not reflected in ERS. Our model performs similarly: 45% of counties require an adjustment to costs in order to match observed planting. Based on this, we believe that the need to adjust costs for our model is a reflection of the discrepancies between real costs and reported costs, rather than a problem mainly caused by our model.

Seventh, their description of crop switching cost is hard to follow. How do they measure the cost of switching? Or do they actually calculate the benefit of switching so that these “switching costs” are actually the benefit to any farmer who does switch?

Thank you for this comment. We interpret the switching costs as additional, hidden costs experienced by farmers. We do not try to measure it, but instead study the effects of a range of costs to understand their effects. These switching costs would reduce the actual switching below the rates predicted by the model without switching costs. We now explicitly say that we “explore the effects of imposing a range of costs in SI 14”, and we have edited the SI section of switching costs to be clearer.

Eighth, the text does not do a good job of describing the results. For example, they do not have a discussion of how crops move in Figure 2 although that is the purpose of the paper.

We will try to respond to these comments one at a time (in each of the responses below). Concerning figure 2, we have developed a new representation of both the new patterns of cultivation and how they are driven by climate change. Our new approach is to show how crops are distributed across temperatures, rather than just across space. This allows us to explain how crops are shifted across the future temperature distribution. We have expanded our discussion of patterns below this figure.

How should one interpret Figure 1 if the crop is not grown in most counties in the map?

These is the predicted responsiveness of crops to high temperatures. We believe that this is a sensible object to estimate even if it covers areas where the actual temperature response is unobserved. Moreover, it is an essential input in the decision-making process for agents considering a change of crops.

What does the top Figure 3 say if the forecast is constrained to keep aggregate acreage (or aggregate yields?) of each crop the same?

We do not think that the constraint to maintain acreage for each crop undermines the information at the top of figure 3, which communicates the extent of crop switching under this constraint. It is clear that considerable crop switching occurs, even under this constraint. The reason for these changes is the same as it would be without the constraint: to take advantage of yield increases and avoid yield decreases. We have added a new SI on the intuition behind our results (SI 11), which we hope adds clarity here.

Does the comparison of actual versus optimal profit in 2010 in bottom of Figure 3 imply that the authors know how to manage every cropland acre better than the farmers that live there?

We certainly do not presume to understand farming better than farmers. We have extended

the discussion of model limitations and added the line, “The cropping patterns shown in our current and future results should not be taken as recommendations, since many details at the field and farmer level are not included.”

In contrast, is the comparison in future years show the difference between assuming farmers do not change crop choice versus they do?

Yes, this is our interpretation, particularly of the lower part of figure 3, where we compare total profits under unchanged planting and under crop switching. There is little doubt that farmers will change crop choices, but since the benefits of this are unclear, we have used our optimization approach to provide some estimates.

They claim but do not show that “extreme degree days [are] the key driver behind climate impacts”. Does EDD explain more than GDD and CID?

Thank you for asking about this. We did not mean that EDD explains more of the variance in yields than GDD or CDI. Most of the variance in yields that is captured for the model is explained by the trend; only for cotton and rice does EDD explain more than GDD and CDI. Our point here was that the emergence of extreme temperatures is a primary vehicle by which climate change is changing yields from what they would be under a counterfactual world without climate change. But this is an empirical statement which we are not in a position to test. We have changed the wording from “the” key driver to “a” key driver.

What is Figure 7 in the Appendix telling the reader?

Variograms are an effective way to display spatial autocorrelation. We have extended the description in the figure caption as follows: “Variance between the coefficients in nearby locations is low (to the left of each graph), but rises as distances increase. The plateau level of the variogram can be compared to the square of coefficient values (which are in the same units), to determine if coefficients vary considerably across space, relative to the average value (as with water deficit coefficients), or if they vary comparatively little (as with technology coefficients).”

Reviewers' comments:

Reviewer #1 (Remarks to the Author):

I have reviewed the responses of the authors to my comments. Many have been address thoughtfully and thoroughly. Others have not. The remaining problems are as follows:

1. Posterior predictive checks have not been performed. Instead, I find a subjective comparison between some data and the predictions of the mean. This is not adequate. It does not show that the model is capable of giving rise to the data. Posterior predictive checks result in a Bayesian P value for test statistics of interest. These checks are covered nicely in all the Bayesian statistics books on my shelf. They are easily computed. No Bayesian model can be assumed to meet fundamental statistical assumptions in their absence.
2. Convergence statistics are not reassuring. Effective sample sizes for the MCMC chains are ridiculously low (< 100) for several quantities. A rule of thumb is that the effective sample size should exceed 5000. There is no excuse for failing to run the simulations with enough iterations to obtain these samples. The authors appear to be reporting them without knowing what they are.
3. Autocorrelation in residuals must be evaluated using a semi-variogram or test for autocorrelation. All inferences are excessively optimistic if residuals that are close to each other in space are more similar than those that are far apart. This is because we can no longer assume the independence that is required for the product likelihood shown in equation 6. There is no reason to assume that the multi-level model solves this problem. If autocorrelation is present, then the spatial structure in residuals must be modeled.
4. The fact that other econometric models have failed to deal with a fundamental assumption of regression modeling, that predictor variables are measured without error, does not excuse to repeat the mistake here.

It appears from these errors that the authors have a somewhat infirm grasp of fundamentals of Bayesian modeling. This reduces my confidence that their inferences are sound.

Reviewer #2 (Remarks to the Author):

The authors have done a great deal to improve the manuscript. It is now much clearer what they are doing and what they have found

Reviewer responses

Thank you for sharing these excellent and useful comments. Please find our responses below.

Reviewer #1 (Remarks to the Author):

1. Posterior predictive checks have not been performed. Instead, I find a subjective comparison between some data and the predictions of the mean. This is not adequate. It does not show that the model is capable of giving rise to the data. Posterior predictive checks result in a Bayesian P value for test statistics of interest. These checks are covered nicely in all the Bayesian statistics books on my shelf. They are easily computed. No Bayesian model can be assumed to meet fundamental statistical assumptions in their absence.

In the revised manuscript we present the Bayesian p-values for two test-quantities — the 10th percentile of the data ($y_{[10]}$) and the range between the 90th percentile and the 10th percentile of the data ($y_{[10]} - y_{[90]}$). As suggested by Gelman et al., [2003], we define the Bayesian p-value as the probability that the replicated data (y^{rep}) could be more extreme than the observed data (y) as measured by these two test quantities. The mean p-value across all the counties is presented for each of the 24 models in section 2.3. The tail area probabilities — $\Pr(T(y^{rep}, \theta) \geq T(y, \theta) | y)$ — in all the cases are within 0.05 and 0.95.

We also present in SI Figure 2, an additional verification of whether the model can reproduce the correlation of observed crop yields with the three climate predictors, i.e., CDI, GDD and EDD for counties that have at least 60 years of data. Three correlation coefficients based on the observed data ($R_{CDI,y}, R_{GDD,y}, R_{EDD,y}$) are computed for each county and compared to the correlation coefficients based on the replicated data ($R_{CDI,y^{rep}}, R_{GDD,y^{rep}}, R_{EDD,y^{rep}}$). The county-wide correlation maps are presented in Figure 2 for all the crops from the models based on full-dataset. We see that the replicated data from the model is also reproducing the correlations well.

2. Convergence statistics are not reassuring. Effective sample sizes for the MCMC chains are ridiculously low (< 100) for several quantities. A rule of thumb is that the effective sample size should exceed 5000. There is no excuse for failing to run the simulations with enough iterations to obtain these samples. The authors appear to be reporting them without knowing what they are.

We ran all the models again using 6 chains and up to 250,000 iterations. The Rhat and the effective sample sizes have improved, and they are all above 1000 now.

3. Autocorrelation in residuals must be evaluated using a semi-variogram or test for autocorrelation. All inferences are excessively optimistic if residuals that are close to each other in space are more similar than those that are far apart. This is because we can no longer assume the independence that is required for the product likelihood shown in equation 6. There is no reason to assume that the multi-level model solves this problem. If autocorrelation is present, then the spatial structure in residuals must be modeled.

We develop a supplemental model to include auto-correlation in the residuals. However, since we found that it is taking an incredibly long time (the MCMC would take an estimated 1.5 million CPU-hours) to estimate this model, we compared the results of the mean log-yield ($\mu_{i,t}$) from the two models (without and with covariance) for a subset of the data, consisting of 100 counties for each crop and found that the bias is within 2%. Based on these checks, and owing to computational constraints, we would like to consider our previous model for the optimal crop choice development scenarios. We include this new model as a supplement in SI Section 7.2 under extended models. Details of the model equations and the results for a subset of 100 counties are provided in this section.

4. The fact that other econometric models have failed to deal with a fundamental assumption of regression modeling, that predictor variables are measured without error, does not excuse to repeat the mistake here.

We appreciate the importance of such errors and we have now performed an analysis which models the true values of all predictors as latent variables, using a Bayesian measurement error model. We have also checked these results using a total least squares estimate (Deming regression).

Two key insights come from this analysis. First, error-in-variables models produce parameter values that are generally more extreme than OLS (or the mode of the equivalent Bayesian regression). This is true for the extreme degree-day coefficient for all crops. This suggests that, indeed, there is an attenuation bias. We believe that this deserves further explanation, but we argue that it is beyond the scope of this paper and that the original model is better suited for our problem. That statement is motivated by the second insight from the analysis: the error-in-variables models are less effective in predicting yields given the original weather data.

This highlights a tension in the paper, between identifying parameter estimates that are most accurate and developing a model for future projection. The goal of the model fitting exercise is to identify spatially-varying marginal effects. This is why we use a relatively simple reduced-form model, rather than representing all of the biological and farming processes that one finds in a process-based model of crop yields. Because of our simplified model, it is not clear whether the latent variable and coefficient differences we find here are due to actual measurement error or model misspecification.

However, for this paper, the predictive capacity of the coefficients of central importance to help us study cropping decision trade-offs in the future. We show in the paper that the Bayesian model, when fit to historically observed measurements, is most effective at predicting future (cross-validated) yields. This is even more true when compared to the error-in-variables models, many of which have negative R^2 values.

The error-in-variables models suggest that our existing estimates of the effects of climate change are conservative. More research is needed to understand the extent of attenuation bias in estimated relationships between weather and yields, but we have kept all of the main results of the paper as those predicted by our original model.

Reviewer #2 (Remarks to the Author):

The authors have done a great deal to improve the manuscript. It is now much clearer what they are doing and what they have found.

Thank you for reviewing our paper and providing the previous helpful comments.

REVIEWERS' COMMENTS:

Reviewer #1 (Remarks to the Author):

I find that the authors responded thoughtfully to my previous comments. I am satisfied with their revisions.